# On Thermal Distribution for Darcy–Forchheimer Flow of Maxwell Sutterby Nanofluids over a Radiated Extending Surface

**DOI:** 10.3390/nano12111834

**Published:** 2022-05-27

**Authors:** Wen Wang, Mohammed M. M. Jaradat, Imran Siddique, Abd Allah A. Mousa, Sohaib Abdal, Zead Mustafa, Hafiz Muhammad Ali

**Affiliations:** 1College of Mechanical Engineering, Xijing University, Xi’an 710123, China; wangwen2227@163.com; 2Mathematics Program, Department of Mathematics, Statstics and Physics, College of Arts and Sciences, Qatar University, Doha P.O. Box 2713, Qatar; zead@qu.edu.qa; 3Department of Mathematics, University of Management and Technology, Lahore 54770, Pakistan; 4Department of Mathematics and Statistics, College of Science, Taif University, P.O. Box 11099, Taif 21944, Saudi Arabia; a.mousa@tu.edu.sa; 5School of Mathematics, Northwest University, No. 229 North Taibai Avenue, Xi’an 710069, China; sohaib@stumail.nwu.edu.cn; 6Department of Mathematics, Khawaja Fareed University of Engineering and Information Technology, Rahim Yar Khan 64200, Pakistan; 7Mechanical Engineering Department, King Fahd University of Petroleum & Minerals, Dhahran 31261, Saudi Arabia; 8Interdisciplinary Research Center for Renewable Energy and Power Systems (IRC-REPS), King Fahd University of Petroleum and Minerals, Dhahran 31261, Saudi Arabia

**Keywords:** Sutterby fluid, Darcy–Forchheimer, electric field, nanofluid, Maxwell fluid, Cattaneo–Christov diffusion

## Abstract

This study addresses thermal transportation associated with dissipated flow of a Maxwell Sutterby nanofluid caused by an elongating surface. The fluid passes across Darcy–Forchheimer sponge medium and it is affected by electromagnetic field applied along the normal surface. Appropriate similarity transforms are employed to convert the controlling partial differential equations into ordinary differential form, which are then resolved numerically with implementation of Runge–Kutta method and shooting approach. The computational analysis for physical insight is attempted for varying inputs of pertinent parameters. The output revealed that the velocity of fluid for shear thickening is slower than that of shear thinning. The fluid temperature increases directly with Eckert number, and parameters of Cattaneo–Christov diffusion, radiation, electric field, magnetic field, Brownian motion and thermophoresis. The Nusselt number explicitly elevated as the values of radiation and Hartmann number, as well as Brownian motion, improved. The nanoparticle volume fraction diminishes against Prandtl number and Lewis number.

## 1. Introduction

The fluids whose viscosity varies non-linearly because of applied stress are referred to as non-Newtonian fluids. Examples of some of these liquids in real-life are ketchup, semen, honey, wax, jellies, etc. Over the last several years, research concerning non-Newtonian fluids has been greatly enhanced owing to their functional effects for technology and manufacturing processes. Analysis on change of viscosity, and thus the behavior of non-Newtonian fluids, was conducted by Shende et al. [1]. The heat transmit by forced convection of a non-Newtonian liquid in a fractional constitutive version boundaries based on a pipe was examined by Chang et al. [2]. For the result of non-Newtonian fluid flow, simulations for the time-lattice for Boltzmann process paradigm were taken by Bisht et al. [3]. Abu et al. [4] analyzed the viscous blend of non-Newtonian fluid in the presence of turbulent flow along with vapor bubble growth. The radiative flow deep research with MHD in signification of non-Newtonian fluids by incorporating the most recent version of the heat flow rate model. By using the modified version of the heat flow rate model, the radiative flowing deep analysis with magnetohydrodynamic on the nature of non-Newtonian fluids was taken by Sohail et al. [5]. Examples of researchers work on non-Newtonoan fluid subject to various types of geometries can be found in [6,7,8,9]. The porous media engagement has significant wide range of applications in the zone of heat transfer design, geothermal, geophysics, under ground water system, recovery system of crude oil, and units of energy storages [10,11]. Ali et al. [12] analyzed the Darcy–Forchheimer medium impacts on the dynamic of nanofluid flow.

The rheological phenomena of non-Newtonian viscosity varying with yields time elastic results pertaining to polymer methods as well as polymer melt. Sutterby fluid, which signifies constitutive equations for extremely polymer aqueous solutions, is one of the most crucial non-Newtonian fluids. To facilitate economic output efficiency, the Sutterby fluid that defines the strictly viscous behavior of the non-Newtonian is known. Scholars are putting their efforts to reveal the properties of Sutterby fluid such as Akram et al. [13], who studded deeply in the presence of electromagnetic fields the Sutterby fluid model which is blood-based graphene oxide nanofluid flows through capillary. Salahuddin et al. [14] examined the dynamics of Sutterby fluid subject to catalytic parabolic surface. Hayat et al. [15] examined the impact of fluid stream from Sutterby and found that it is prone to homogeneous-heterogeneous as well as nonlinear radiation transformations. Nawaz et al. [16] discussed the function of hybrid nanoparticles throughout the thermal efficiency of ethylene glycol, the Sutterby fluid. Thermal and energy stratified flow analysis of Sutterby nanofluid with zero mass flux status was taken by Mir et al. [17]. Sabir et al. [18] examined the consequences of heat radiation and inclined magnetic force on the Sutterby fluid employing the Cattaneo–Christov thermal gradient scheme.

Nanofluids are possible heat storage liquids with enhanced thermos-physical characteristics, so analytical platforms for optimized illustrations may be associated through heat trading activity. Study nowadays in the domain of nanomaterials has quickly evolved influencing to its extensive deployments in various areas, due to its wide range uses, such as thermal transportation. Many intellectuals have paid too much attention to the new aspects in this domain. Mahdavi et al. [19] extensively reviewed nanofluid jet refrigerating fluid motion and heat transition assessment on a hot exterior with variable roughness. Stability analysis of nanofluids was carried out by Chakraborty et al. [20]. Esfe et al. [21] explored nanofluids streamline for enhanced oil recovery in kind of a heterogeneous two-dimensional anticline geometry. Swaim et al. [22] demonstrated a comprehensive inspection into the impacts of accelerating heating source across an inclined magnetic flux. Many researchers also explored nanofluids in different areas [23,24,25,26,27].

It is possible to find strong energy throughout the domain of nanofluids. They can also be used to determine the effects of calming stress. A description of fluids of the pressure pattern, viz., the model of Maxwell will predict stress relief and has become much more general instead. The research of Maxwell flow of nanofluid has grown substantially in latest generations leading to several applications in engineering including medical processes. Sharma et al. [28] numerically studied the Maxwell nanofluid graphene flows past a uniformly stretched sheet. Maxwell base fluid stream including magnetohydrodynamic dissipative and radiative graphene was taken by Hussain et al. [29]. Abro et al. [30] analyzed the thermophysical characteristics of Maxwell nanofluids with normal kernel through fractional derivatives. A mathematical and statistical approach to the effect of radiative heat flux in Maxwell flow of viscoastic fluid over a chemically reacted spiraling disc was carried out by Ahmad et al. [31]. Microstructure-like substantial flowing along with bio-convection with inertial properties of Magnetohydrodynamic suspended SWCNT- and MWCNT-dependent Maxwell nanofluid examined by Shah et al. [32]. Ali et al. [33] reviewed gyrotactic microorganisms Falkner–Skan flow of Maxwell nanofluid with activation energy above a wedge.

From the above literature the authors discovered that the thermal distribution for transportation of Darcy–Forchheimer Maxwell Sutterby nanofluid flow in the availability of Cattaneo–Christov heat transition and electromagnetic field is rarely taken into account. Furthermore, the use of convective boundary conditions and bio-convection of microbes enhances to the innovation of this research. Because of the pairing and higher order non-linearity of the governing boundary value problem, arithmetical findings are achieved by conducting the Runge–Kuttta strategy code with shooting notion on the Matlab program. The findings are affirmed as a specific situation of prior findings.

## 2. Mathematical Formulation

By considering dissipated Darcy–Forchheimer multi-slip constraints over a time-independent flow of Maxwell Sutterby nanofluid. A two-dimensional electro- and magnetohydrodynamic boundary with the presence of Lorentz force [34] and incompressible Maxwell Sutterby fluid is defined as $F=\sigma_{m}\left(E_{f}+\left(J_{1}*B_{0}\right)\right)$.

A base is extended with velocity $U_{w}\left(x\right)=a_{1}x$ along boundary. On the surface of the layer, a constant $T_{w}$ temperature is provided in vertical direction, a magnetic flux of intensity $B_{0}$ is imposed upon this flow (see Figure 1). According to the above-mentioned assumption, the governing equations are [35,36,37]
(1)$$\begin{array}{c} \frac{\partial u}{\partial x}+\frac{\partial v}{\partial y}=0, \end{array}$$
(2)$$\begin{array}{c} u\frac{\partial u}{\partial x}+v\frac{\partial u}{\partial y}=\frac{\nu }{2}\frac{\partial^{2}u}{\partial y^{2}}\left(1+\frac{Sb_{c}^{2}}{2}{\left(\frac{\partial u}{\partial y}\right)}^{2}\right)+\frac{\sigma_{m}B_{0}}{\rho }\left(E_{f}-B_{0}u\right)-\frac{\nu }{k^{\prime }}u\\ -\lambda_{1}\left[u^{2}\frac{\partial^{2}u}{\partial x^{2}}+v^{2}\frac{\partial^{2}u}{\partial y^{2}}+2uv\frac{\partial^{2}u}{\partial y\partial x}\right]-\frac{C_{F}^{*}\sqrt{k^{\prime }}}{\nu }u^{2}, \end{array}$$
(3)$$\begin{array}{c} u\frac{\partial T}{\partial x}+v\frac{\partial T}{\partial y}=\frac{k_{1}}{\rho C_{p}}\frac{\partial^{2}T}{\partial y^{2}}+\frac{\rho_{p}C_{p}}{\rho C}\left[D_{B}\frac{\partial C}{\partial y}\frac{\partial T}{\partial y}+\frac{D_{T}}{T_{\infty }}{\left(\frac{\partial T}{\partial y}\right)}^{2}\right]\\ +\frac{\mu }{\rho C_{p}}{\left(\frac{\partial u}{\partial 2}\right)}^{2}-\frac{1}{\rho C_{p}}\frac{\partial q_{r}}{\partial y}+\frac{\sigma_{2}}{\rho C_{p}}{\left(uB_{0}-E_{f}\right)}^{2}-\tau_{1}B^{*}, \end{array}$$
(4)$$\begin{array}{c} u\frac{\partial C}{\partial x}+v\frac{\partial C}{\partial y}=D_{B}\frac{\partial }{\partial y}\frac{\partial C}{\partial y}+\frac{D_{T}}{T_{\infty }}\frac{\partial }{\partial y}\frac{\partial T}{\partial y}. \end{array}$$
$$B^{*}=\left[u\frac{\partial u}{\partial x}\frac{\partial T}{\partial x}+v\frac{\partial v}{\partial x}\frac{\partial T}{\partial x}+u\frac{\partial v}{\partial x}\frac{\partial T}{\partial x}+v\frac{\partial u}{\partial y}\frac{\partial T}{\partial x}+2uv\frac{\partial^{2}T}{\partial x\partial y}+u^{2}\frac{\partial }{\partial x}\frac{\partial T}{\partial x}+v^{2}\frac{\partial }{\partial y}\frac{\partial T}{\partial y}\right].$$

Here, *u* and *v* represent the component of velocity in *x*- and *y*-direction, respectively; the flow deportment index is *S*; $b_{c}^{2}$ represents consistency index; $\sigma_{m}$ is the magnetic permeability; ρ signifies density; electric field strength is $E_{f}$; $C_{F}^{*}$ symbolizes Forchheimer quantity; *T* represents temperature; *C* indicates solutal density; $k_{1}$ represents thermal conductivity; $C_{p}$ signifies specific heat; $\tau_{1}$ is heat relaxation time; $q_{r}=\left(\frac{-4T_{\infty }^{3}\sigma_{m}}{3k^{**}}\right)\frac{\partial }{\partial y}T^{4}$ denotes radiative heat flux; *C* represents nanoparticles concentration; and Brownian motion constant is viewed by $D_{B}$ and the thermophorsis constant is expressed by $D_{T}$.

The model’s boundary situations are given below.
(5)$$\begin{array}{c} \left.\begin{array}{c} u=U_{w}+N_{0}\frac{\partial u}{\partial y},v=0,T=T_{w}\left(x\right)+k_{0}\frac{\partial T}{\partial y},C=C_{w}\left(x\right)+k_{2}\frac{\partial C}{\partial y},\;as\;y=0,\\ u\to 0,\;T\to T_{\infty },\;C\to C_{\infty },\;as\;y\to \infty . \end{array}\right\} \end{array}$$

Here, $U_{w}=a_{1}x,a_{1}>0$ denotes stretching velocity, $T_{\infty }$ symbolizes ambient temperature, slip length is $N_{0}$, $C_{w}$ designates wall concentration, $C_{\infty }$ denotes ambient concentration, and $k_{0}$ signifies heat jump length and $k_{2}$ depicts the density jump length.

Using the similarity variables [33]
(6)$$\begin{array}{c} \eta =\sqrt{\frac{a}{\nu }}y,\psi =\sqrt{a\nu }xf\left(\eta \right),\;\theta \left(\eta \right)=\frac{T-T_{\infty }}{T_{w}-T_{\infty }},\;\phi \left(\eta \right)=\frac{C-C_{\infty }}{C_{w}-C_{\infty }}. \end{array}$$

Substituting relation (6) in Equations (2)–(4), we get
(7)$$\begin{array}{c} f^{\prime \prime \prime }-2f^{\prime 2}+2ff^{\prime \prime }-\beta (f^{2}f^{\prime \prime \prime }-2ff^{\prime }f^{\prime \prime })-\frac{S}{2}ReDef^{\prime \prime 2}f^{\prime \prime \prime }+2Ha^{2}\left(E_{1}-f^{\prime }\right)-2K_{p}f^{\prime }-2F_{1}^{*}f^{\prime 2}=0, \end{array}$$
(8)$$\begin{array}{c} \frac{1}{Pr}\left(1+\frac{4}{3}Rd\right)\theta^{\prime \prime }+f\theta^{\prime }+Nb\theta^{\prime }\phi^{\prime }+Nt\theta^{\prime 2}-b\left(f^{2}\theta^{\prime \prime }+ff^{\prime }\theta^{\prime }\right)+Ecf^{\prime \prime 2}+EcHa^{2}\left(f^{\prime }-E_{1}\right)=0, \end{array}$$
(9)$$\begin{array}{c} \phi^{\prime \prime }+PrLef\phi^{\prime }+\frac{Nt}{Nb}\theta^{\prime \prime }=0. \end{array}$$
where transformed boundary conditions (5),
(10)$$\begin{array}{c} \left.\begin{array}{c} f\left(0\right)=0,\;f^{\prime }\left(0\right)=1+\delta f^{\prime \prime },\;\theta \left(0\right)=1+\beta \theta^{\prime },\;\phi \left(0\right)=1+\gamma \phi^{\prime },\;at\;\eta =0,\\ f^{\prime }\left(\infty \right)\to 0,\;\theta \left(\infty \right)\to 0,\phi \left(\infty \right)\to 0,\;as\;\eta \to \infty . \end{array}\right\} \end{array}$$

The non-dimensional factors are listed in the preceding order:

β=λa is Maxwell fluid Deborah number, $Ha=\sqrt{\frac{\sigma_{m}}{\rho }}B_{0}$ is the magnetic parameter, $Re=\frac{ax^{2}}{\nu }$ is the Reynolds number, Deborah number is $De=b^{2}a^{2}$, electric parameter is $E_{1}=\frac{E_{f}}{xaB_{0}}$, porosity parameter is $Kp=\frac{\nu }{k^{\prime }a}$, inertia parameter is $F_{1}^{*}=\frac{C_{F}^{*}}{\sqrt{k^{\prime }}}$, Prandtl number is $Pr=\frac{\nu }{\alpha }$, radiation parameter is $Rd=\frac{16T_{\infty }^{3}\sigma_{m}}{3k^{**}\kappa }$, $Ec=\frac{a^{2}x^{2}}{C_{p}\left(T_{w}-T_{\infty }\right)}$ denotes Eckert number, $b=\tau_{1}a$ is the constant of thermal relaxation, $Nb=\frac{\rho_{p}C_{p}}{\alpha \rho C}D_{B}\left(C_{w}-C_{\infty }\right)$ and $Nt=\frac{\rho_{p}C_{p}D_{T}\left(T_{w}-T_{\infty }\right)}{\alpha \rho CT_{\infty }}$ are Brownian motion and thermophoresis parameter, Lewis number is $Le=\frac{\nu }{D_{B}}$.

The physical quantities are signified as follows [38]:

$Cf_{x}$ (skin friction coefficient), $Nu_{x}$ (local Nusselt number), and $Sh_{x}$ (local Sherwood number) are given below:$$Cf_{x}=\frac{\tau_{w}}{\rho U^{2}w},\;Nu_{x}=\frac{xq_{w}}{k_{1}\left(T_{w}-T_{\infty }\right)},\;Sh_{x}=\frac{xq_{m}}{D_{B}\left(C_{w}-C_{\infty }\right)},$$
where $\tau_{w}$, $q_{w}$, and $q_{m}$ denotes shear stress, surface heat flux and surface mass flux are given by (at y=0),
$$\tau_{w}=-\mu \left[\left(1+\beta \right)\frac{\partial u}{\partial y}+\frac{Sb_{c}^{2}}{3}{\left(\frac{\partial u}{\partial y}\right)}^{3}\right],\;q_{w}=-k_{1}\frac{\partial T}{\partial y},\;q_{m}=-D_{B}\frac{\partial C}{\partial y}$$

When we solve these quantities utilizing the assigned similarity transformation, we get
$$C_{f}{\left(Re_{x}\right)}^{-1/2}=-\left[\left(1+\beta \right)\left(f^{\prime \prime }\left(0\right)+\frac{S}{3}ReDef^{\prime \prime }{\left(0\right)}^{3}\right)\right],\;Nu_{x}{\left(Re_{x}\right)}^{-1/2}=-\theta^{\prime }\left(0\right),\;Sh_{x}{\left(Re_{x}\right)}^{-1/2}=-\phi^{\prime }\left(0\right),$$
where, $\left(Re_{x}\right)=\frac{xU_{w}}{\nu }$ signifies the local Reynolds number.

## 3. Solution Procedure

Numerical scheme is coded in Matlab software to get the graphical and tabular output. First-order scheme with just some factor implemented as shown below [39,40,41,42]:$$\begin{array}{c} s_{1}^{\prime }=s_{2}\\ s_{2}^{\prime }=s_{3}\\ s_{3}^{\prime }=\left(-1\right)\left[-2s_{2}^{2}+2s_{1}s_{3}-\beta \left(s_{1}^{2}ds_{3}-2s_{1}s_{2}s_{3}\right)-\frac{S}{2}ReDes_{3}^{2}s_{3}^{\prime }+2Ha^{2}\left(E_{1}-s_{2}\right)-2K_{p}s_{2}-2F_{1}^{*}s_{2}^{2}\right]\\ s_{4}^{\prime }=s_{5}\\ s_{5}^{\prime }\left(\frac{1}{Pr}\left(1+\frac{4}{3}Rd\right)\right)=\left(-1\right)\left[s_{1}s_{5}+Nbs_{5}s_{7}+Nts_{5}^{2}-b\left(s_{1}^{2}\theta^{\prime \prime }+s_{1}s_{2}s_{5}\right)+Ecs_{3}^{2}+EcHa^{2}\left(s_{2}-E_{1}\right)\right]\\ s_{6}^{\prime }=s_{7}\\ s_{7}^{\prime }=-PrLes_{1}s_{7}-\frac{Nt}{Nb}ds_{5} \end{array}$$
along with the boundary conditions:$$\begin{array}{c} s_{1}=0,\;s_{2}=1+\delta s_{2},\;s_{3}=h,\;s_{4}=1+\beta s_{4},\;s_{5}=g,\;s_{6}=1+\gamma s_{6},\;s_{7}=i,\mathrm{at}\;\eta =0\\ s_{2}\to 0,\;s_{4}\to 0,\;s_{6}\to 0\;as\;\eta \to \infty . \end{array}$$

The unknown initial conditions $s_{3}$, $s_{5}$, $s_{7}$, are allotted arbitrary values to begin the computational methodology once the solution validates the boundary conditions, these values are finalized.

## 4. Results and Discussion

The numerical procedure, as described in the above section, yielded a solution of the controlling equations. The dependent physical variables like temperature of fluid θ(η), velocity $f^{\prime }\left(\eta \right)$, nanoparticle volume fraction ϕ(η), Nusselt number $-\theta^{\prime }\left(0\right)$, skin friction factor $-f^{\prime \prime }\left(0\right)$ and Sherwood number $-\phi^{\prime }\left(0\right)$. For appropriate variations of influential factors, the variable attitude of these parameters has been calculated. The current arithmetic coding is ascertained because there appears to be a satisfactory agreement between the current and previously existing outcomes (see Table 1).

The graphs in Figure 2, Figure 3, Figure 4, Figure 5, Figure 6, Figure 7, Figure 8, Figure 9 and Figure 10 are sketched for two cases of Sutterby fluid parameter *S* (S=−0.5,S=0.5). Note that S>0 is related to shear thickening and S<0 indicates shear thinning. Note that the velocity $f^{\prime }\left(\eta \right)$ for shear thinning is faster than that for shear thickening. Figure 2 displays the influence of Maxwell parameter β and electric parameter $E_{1}$ on $-f^{\prime }\left(\eta \right)$. Both of these parameters have the potential to increase the flow speed. The impacts of porosity parameter Kp, Hartmann number Ha and inertia parameter $F_{1}^{*}$ on velocity $f^{\prime }\left(\eta \right)$ are plotted in Figure 3. It is reviewed that the flow become faster when Ha is slightly intensified but it decelerates against the augmented values of Kp and $F_{1}^{*}$. Physically, the existence of the resistive force in the form of Lorentz force is due to the inclusion of enhancing external magnetic field and leads to deceleration of the velocity, but an opposite behavior is perceived for temperature distribution, and the porous medium interaction makes the fluid more viscous, which slows down the velocity. Figure 4 presents the varying pattern of fluid velocity when Deborah number De and Reynolds number Re are enhanced. Note that these parameters directly increase the velocity $f^{\prime }\left(\eta \right)$ for shear thinning but retard the flow for shear thickening. The viscoelastic effects generate the resistance force which cause the profile of velocity to decline. Figure 5 displays a lowering of the fluid temperature θ(η) against Pr and an increase in θ(η) when radiation parameter Rd is promoted in value. The results in this figure can be explained on the basis of physical nature of the parameters Pr and Rd. The Prandtl number is related inversely to thermal diffusivity, its higher values are responsible for the diminishing temperature distribution. The large inputs of radiation parameter Rd means incremented radiative mode of heat and hence rise in temperature is attained.

From Figure 6, it is depicted that the temperature increases when Nb and Nt increase. The faster Brownian movement and thermophorsis strengthen the thermal distribution. The nanoparticles traveled from hot region to cold region due to thermophoretic force, and heat transfer rate is increased at the boundary surface. Similarly, faster movement of tiny particles raised the Brownian force, which boosts the base fluid temperature. In Figure 7, the plots for boosted temperature distribution are delineated with growing inputs of Cattaneo–Christov parameter b and Hartmann number Ha. A similar improved pattern of θ(η) are sketched in Figure 8, when Eckert number Ec and electric parameter $E_{1}$ are enhanced. The nanoparticle volume fraction ϕ(η) diminishes against Nb, but it enhances with higher values of Nt as noticed from Figure 9. Figure 10 reveal that the nanoparticle volume fraction ϕ(η) declines significantly against the growing values of Pr and Le. Table 2 identifies the skin friction coefficient −f″(0), which declines with expanding Hartmann number Ha as well as Maxwell fluid factor β and although intensifies effectively with *S*, Re, De, Kp and $F_{1}^{*}$. The electric parameter $E_{1}$ does not make any mentionable effects on $-f^{\prime \prime }\left(0\right)$. The Nusselt quantity $-\theta^{\prime }\left(0\right)$ enhances as the factors Rd, Ha as well as Nb are raised, but lowers when the variables Pr, *b*, Ec, $E_{1}$ and Nt are expanded, as revealed in Table 3. Table 4 portrays the accelerated advancement of the Sherwood quantity $-\phi^{\prime }\left(0\right)$ if the values of Le as well as Nt are improved, and it lessens if the values of Pr as well as Nb are expanded.

## 5. Conclusions

The electro-magnetohydrodynamic boundary layer transport of Maxwell Sutterby nanofluid with multi-slip conditions across an extending sheet is explored. A brief description of the significant findings is as follows:The velocity gradient enhanced as the magnitude of β, $E_{1}$ and Ha elevated, although it dropped significantly as the valuation of Kp and $F_{1}^{*}$ extended.It is worth mentioning that the velocity field uplifted for De and Re when Sutterby parameter *S* take negative values and decrease when Sutterby parameter is positive.Temperature goes up if the magnitudes of Rd, Nb, Nt, *b*, Ha, Ec as well as $E_{1}$ upsurge, whereas it drops as the valuation of Pr grows.The concentration distribution for Nb, Pr and Le reveals a declining trend while upsurge with higher Nt.Skin friction reduces when Ha and β takes higher values. Furthermore, it escalates in direct proportion to *S*, 4Re, De, kp and $F_{1}^{*}$.The Nusselt number explicitly elevated as the values of Rd, Ha as well as Nb improved. The inverse attitude of Pr, *b*, Ec, $E_{1}$ and Nt is revealed.The Sherwood quantity falls for Pr as well as Nb while surging for Le and Nt.

## Figures and Tables

**Figure 1 nanomaterials-12-01834-f001:**
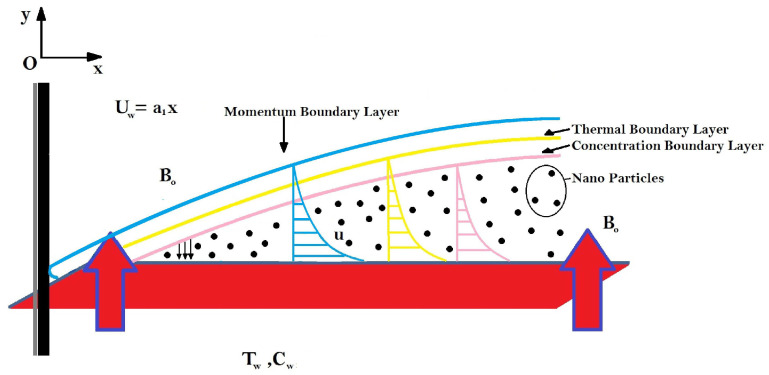
Problem description.

**Figure 2 nanomaterials-12-01834-f002:**
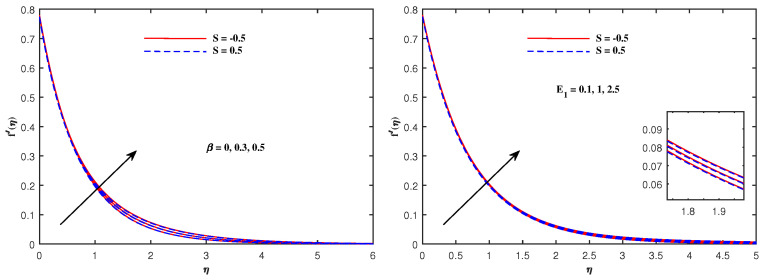
Velocity variation with β and $E_{1}$.

**Figure 3 nanomaterials-12-01834-f003:**
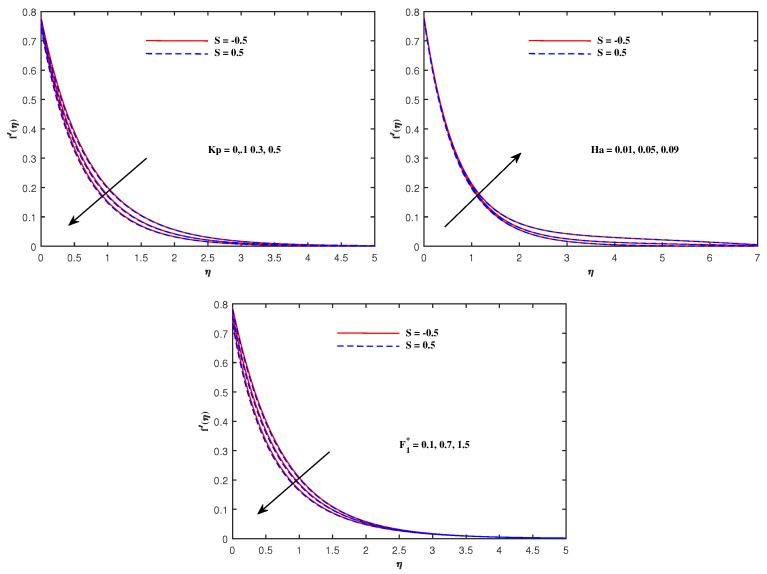
Velocity variation with Kp, Ha and $F_{1}^{*}$.

**Figure 4 nanomaterials-12-01834-f004:**
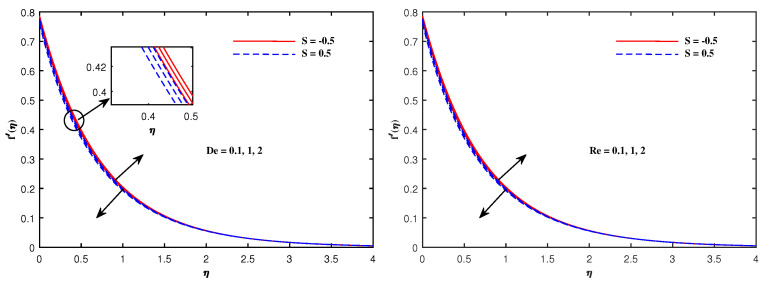
Velocity variation with De and Re.

**Figure 5 nanomaterials-12-01834-f005:**
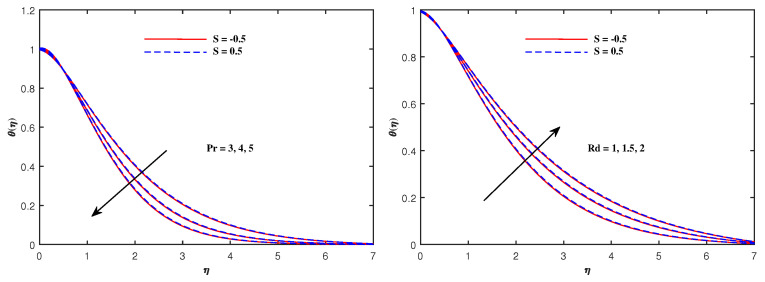
Temperature variation with Pr and Rd.

**Figure 6 nanomaterials-12-01834-f006:**
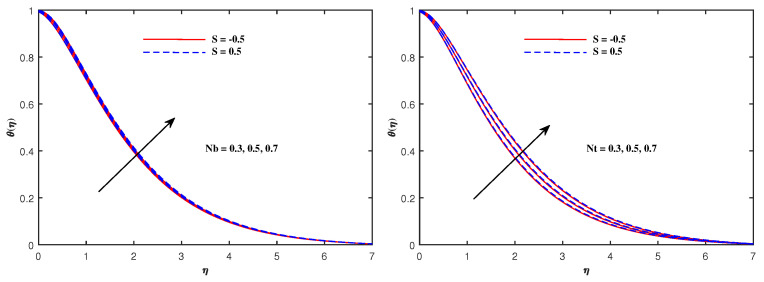
Temperature variation with Nb and Nt.

**Figure 7 nanomaterials-12-01834-f007:**
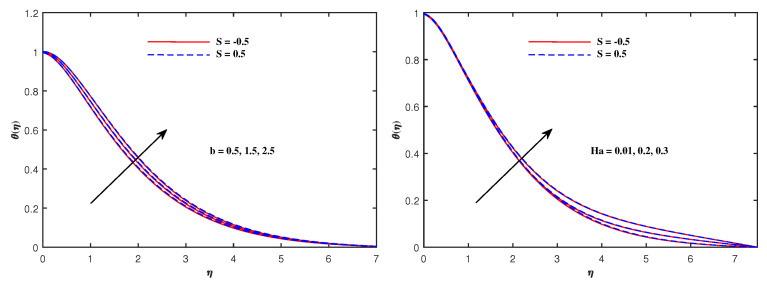
Temperature variation with *b* and Ha.

**Figure 8 nanomaterials-12-01834-f008:**
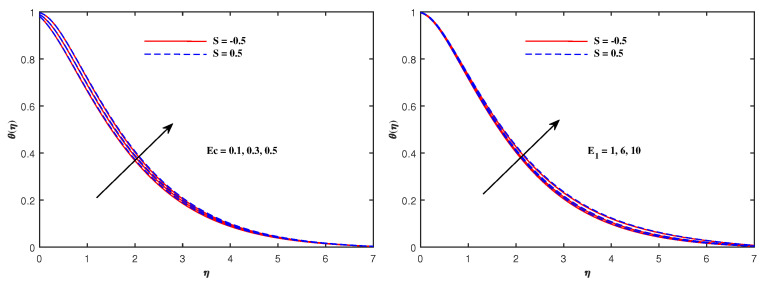
Temperature variation with Ec and $E_{1}$.

**Figure 9 nanomaterials-12-01834-f009:**
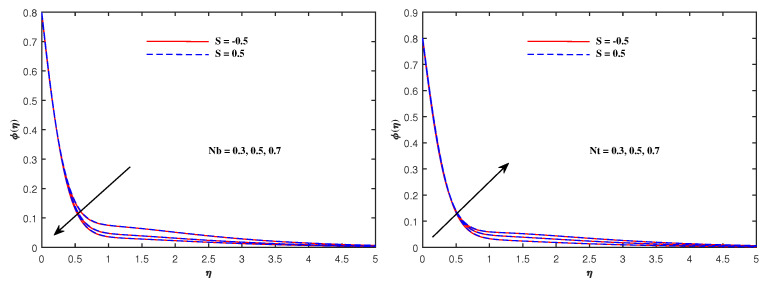
Concentration variation with Nb and Nt.

**Figure 10 nanomaterials-12-01834-f010:**
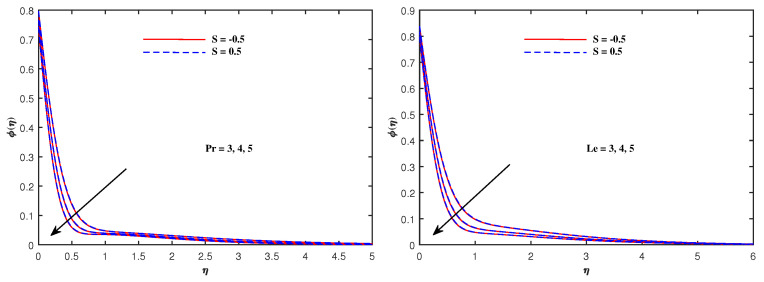
Concentration variation with Pr and Le.

**Table 1 nanomaterials-12-01834-t001:** The comparative outputs for $-f^{\prime \prime }\left(0\right)$.

Ha	Ibrahim and Negera [35]	Sajid et al. [38]	Present Results
0.0	1.2105	1.1706	1.1917
0.3	1.3578	1.3393	1.3485
0.5	1.4478	1.4408	1.4456
1.0	1.6504	1.6677	1.6545

**Table 2 nanomaterials-12-01834-t002:** Results for $-f^{\prime \prime }\left(0\right)$.

Ha	*S*	Re	De	β	$E_{1}$	Kp	$F_{1}^{*}$	$-f^{\prime \prime }\left(0\right)$
0.01	0.5	0.5	0.5	0.1	1.0	0.1	0.3	1.0792
0.03								1.0780
0.05								1.0758
0.01	0.1							1.0191
	0.3							1.0486
	0.5							1.0792
	0.5	0.5						1.0792
		0.7						1.1109
		0.9						1.1440
		0.5	0.5					1.0792
			1.0					1.1610
			1.5					1.2522
			0.5	0.0				1.1889
				0.1				1.0792
				0.2				0.9685
				0.1	1.0			1.0792
					2.0			1.0790
					3.0			1.0788
					1.0	0.1		1.0792
						0.2		1.1209
						0.3		1.1608
						0.1	0.1	1.0359
							0.2	1.0579
							0.3	1.0792

**Table 3 nanomaterials-12-01834-t003:** Results for $-\theta^{\prime }\left(0\right)$.

Rd	Pr	*b*	Ec	Ha	$E_{1}$	Nt	Nb	$-\theta^{\prime }\left(0\right)$
1.0	3.0	0.5	0.5	0.01	1.0	0.5	0.5	0.0408
2.0								0.0831
3.0								0.0985
1.0	1.0							0.1063
	2.0							0.0773
	3.0							0.0408
	3.0	0.1						0.0517
		0.3						0.0462
		0.5						0.0408
		0.5	0.1					0.1875
			0.3					0.1145
			0.5					0.0408
			0.5	0.01				0.0408
				0.03				0.0416
				0.05				0.0429
				0.01	1.0			0.0408
					3.0			0.0403
					5.0			0.0387
					1.0	0.1		0.0921
						0.3		0.0647
						0.5		0.0408
						0.5	0.1	0.1075
							0.3	0.0709
							0.5	0.0408

**Table 4 nanomaterials-12-01834-t004:** Results for $-\phi^{\prime }\left(0\right)$.

Le	Pr	Nb	Nt	$-\phi^{\prime }\left(0\right)$
3.0	3.0	0.5	0.5	1.6124
4.0				1.8410
5.0				2.0313
5.0	1.0			1.1343
	2.0			1.6545
	3.0			2.0313
	3.0	0.1		2.2199
		0.3		2.0662
		0.5		2.0313
		0.5	0.1	1.9574
			0.3	1.9925
			0.5	2.0313

## Data Availability

Not applicable.

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
