# Peer review of "On Thermal Distribution for Darcy–Forchheimer Flow of Maxwell Sutterby Nanofluids over a Radiated Extending Surface"

_nanomaterials, 2022, doi:10.3390/nano12111834_

Round 1
Reviewer 1 Report
This paper addresses thermal transportation associated with the dissipated flow of Maxwell Sutterby nanofluid caused by an elongating surface. The analysis was performed by computational analysis. The obtained findings are interesting and important for industrial application. The reviewer proposes this paper can be accepted after following minor revisions.
- All the results can be qualitatively predictable? What is the most optimal condition? Some quantitative discussion is required.
- If possible, velocity and temperature profiles should be formulated by the use of some non-dimensional parameters.
- The important findings in the abstract should be matched to that in conclusion.
- The table for all nomenclatures is required in the text.
Author Response
Reviewer: 1
Response:
Dear respected Reviewer,
We are thankful to you for your positive comments to improve our manuscript.
We revised the manuscript according to the points and highlights the changes with color. Please
look at the revised version
Reviewer Comments:
This paper addresses thermal transportation associated with the dissipated flow of Maxwell Sutterby
nanofluid caused by an elongating surface. The analysis was performed by computational
analysis. The obtained findings are interesting and important for industrial application. The
reviewer proposes this paper can be accepted after following minor revisions.
1. All the results can be qualitatively predictable? What is the most optimal condition? Some
quantitative discussion is required.
Response:
Done. Please look at the revised manuscript results and discussion section. We update the results
and discussion section according to the wise suggestion.
2. If possible, velocity and temperature profiles should be formulated by the use of some non-dimensional parameters.
Response:
We used non-dimensional parameters mentioned in similarity transformations equation no 6 to
convert velocity and temperature profiles to dimensionless form. Please see equation no 6.
The important findings in the abstract should be matched to that in conclusion.
Response:
Done. Please look at the revised manuscript abstract.
3. The table for all nomenclatures is required in the text.
Response:
Done. Please look at the revised manuscript.

Reviewer 2 Report
This study is concerned with the thermal transportation associated with the dissipated flow of Maxwell Sutterby nanofluid caused by an elongating surface within a Darcy-Forchheimer sponge medium. Runge-Kutta method and shooting approach are applied to simulate the nano field. From the outputs, it is observed that the velocity of fluid for shear thickening is slower than that of shear thinning, and the fluid temperatures rise directly with physical parameters.
Overall, the work has merits in the publication in the Journal of Nanomaterials. However, the following issues should be clarified beforehand:
A) – Some typographical and grammatical mistakes should be corrected, with close language editing. “slower then”!, “literature the authors comes to know”!
B) – The significance of Darcy-Forchheimer Maxwell Sutterby nanofluid flow should be given with potential applications.
C) - In the boundary conditions (8), it is assumed that the Riga plate is stretched. How can it be possible? Explain.
D) - The literature should be updated with more recent publications such as “A flow behavior of Sutterby nanofluid near the catalytic parabolic surface (DOI: 10.1016/j.icheatmasstransfer.2021.105821)”, “Numerical study of nanofluid flow and heat transfer over a rotating disk using Buongiorno's model (DOI: 10.1108/HFF-08-2015-0328)”, “Analysis of thermally stratified flow of Sutterby nanofluid with zero mass flux condition (DOI: 10.1016/j.jmrt.2019.11.088)” and “Heat Transfer Enhancement Feature of the Non-Fourier Cattaneo-Christov Heat Flux Model (DOI: 10.1115/1.4051671)”.
E) – Model equations (1-5) require referencing. Also, the novel terms studied here should be highlighted.
F) – Is the reduced model in Eqs.(7-10) locally similar? If so, explicitly state that. Can it be solvable analytically?
G) – From Table 1, some differences are observed from the published data, why?
H)- Rephrase “Skin fraction reduces”.
Author Response
Reviewer: 2
Response:
Dear respected Reviewer,
We are thankful to you for your positive comments to improve our manuscript.
We revised the manuscript according to the points and highlights the changes with color. Please
look at the revised version
Reviewer Comments:
This study is concerned with the thermal transportation associated with the dissipated flow of Maxwell
Sutterby nanofluid is caused by an elongating surface within a Darcy-Forchheimer sponge medium.
Runge-Kutta method and shooting approach are applied to simulate the nanofield. From the
outputs, it is observed that the velocity of fluid for shear thickening is slower than that of shear
thinning, and the fluid temperatures rise directly with physical parameters. Overall, the work has
merits in the publication in the Journal of Nanomaterials. However, the following issues should be
clarified beforehand:
A) – Some typographical and grammatical mistakes should be corrected, with close language
editing. “slower then”!, “literature the authors comes to know”!
Response:
Done. Please look at the revised manuscript.
B) – The significance of Darcy-Forchheimer Maxwell Sutterby nanofluid flow should be given
with potential applications.
Response:
Done. Please look at the revised manuscript introduction section.
C) - In the boundary conditions (8), it is assumed that the Riga plate is stretched. How can it be
possible? Explain.
Response:
Actually, we considered stretching plate and that’s why the Riga plate is stretched. Please see the
below references.
[1] Transient rotating nanofluid flow over a Riga plate with gyrotactic micro-organisms, binary
chemical reaction and non-Fourier heat flux
[2] EMBL Nano Fluid Flow Along Riga Plate in a Rotating System.
D) - The literature should be updated with more recent publications such as “A flow behavior of
Sutterby nanofluid near the catalytic parabolic surface (DOI:
10.1016/j.icheatmasstransfer.2021.105821)”, “Numerical study of nanofluid flow and heat
transfer over a rotating disk using Buongiorno's model (DOI: 10.1108/HFF -08-2015-0328)”,
“Analysis of thermally stratified flow of Sutterby nanofluid with zero mass flux condition (DOI:
10.1016/j.jmrt.2019.11.088)” and “Heat Transfer Enhancement Feature of the Non-Fourier
Cattaneo-Christov Heat Flux Model (DOI: 10.1115/1.4051671)”
Response:
Done. Please see the revised manuscript.
E) – Model equations (1-5) require referencing. Also, the novel terms studied here should be
highlighted.
Response:
Done. Please look at the revised manuscript.
F) – Is the reduced model in Eqs.(7-10) locally similar? If so, explicitly state that. Can it be solvable
analytically?
Response:
Actually, the obtained systems of equations are highly non-linear so it is very difficult to
obtained analytical solution.
G) – From Table 1, some differences are observed from the published data, why?
Response:
Our results have an excellent correlation with Ibrahim et al. and Sajid et al. If we see the results
of Ibrahim against Ha = 0.0 are 1.2105 and Sajid et al. are 1.1706. The difference between
Ibrahim et al.and Sajid et al. is 0.0399, but if we compare our results with Ibrahim et al. then the
difference is less as compared to Sajid et al. Like only 0.0188.
Ha Ibrahim and Negera [27] Sajid et al. [26] Present Results
0.0 1.2105 1.1706 1.1917
0.3 1.3578 1.3393 1.3485
0.5 1.4478 1.4408 1.4456
1.0 1.6504 1.6677 1.6545
H)- Rephrase “Skin fraction reduces”.
Response:
Done. Please see the revised manuscript.